# How fake news can turn against its spreader

**Dorje C. Brody** [1,2,3], **Tomooki Yuasa** [4]*

**1** School of Mathematics and Physics, University of Surrey, Guildford, United Kingdom, **2** Department of Mathematics, Imperial College London, London, United Kingdom, **3** Institute of Industrial Science, The University of Tokyo, Tokyo, Japan, **4** Faculty of Economics and Business Administration, Graduate School of Management, Tokyo Metropolitan University, Tokyo, Japan

\* tomooki-yuasa@tmu.ac.jp

## Abstract

When different information sources on a given topic are combined, they interact in a non-trivial manner for a rational receiver of these information sources. Suppose that there are two information sources, one is genuine and the other contains disinformation. It is shown that under the conditions that the signal-to-noise ratio of the genuine information source is sufficiently large, and that the noise terms in the two information sources are positively correlated, the effect of disinformation is reversed from its original intent. That is, the effect of disinformation on a receiver of both information sources, who is unaware of the existence of disinformation, is to generate an opposite interpretation. While the condition in which this phenomenon occurs cannot always be ensured, when it is satisfied, the effect provides an effective way of countering the impacts of disinformation.

## Introduction—Modelling disinformation

While the concept of "fake news" in one form of another has been around for centuries, the impact of disinformation on political stages worldwide has gained unprecedented attention over the past decade [1–9]. To assess and understand the impact of disinformation on democratic processes, it will be useful to develop generative models that can be used to simulate different scenarios on how people's perceptions on a given topic change in accordance with the revelation of information about that topic. An important point to note here is the fact that any such information-driven model has to fall squarely within the framework of communication theory. The reason is simple. Whether a given message contains disinformation or not, communication theory is designed to analyse how information encoded in that message is processed by the receiver [10].

In communication theory, information contained in a message is decomposed into its signal component and noise component, while the relative magnitudes of the two is measured by the signal-to-noise ratio [11]. Exactly how signal and noise should be combined depends on the application, but if the noise is normally distributed, then it is natural to combine them in an additive manner, that is, we have a signal-plus-noise decomposition. In the simplest case, if the signal—part of the message that one wishes to know—is modelled as a random variable $X$, and if noise—part of the message that is independent of the signal—is modelled likewise as

**Data availability statement:** All relevant data are within the manuscript and its Supporting information files.

**Funding:** This work was supported by JSPS KAKENHI (Grant Number: 22K13965 to T.Y.). The funder is the Japan Society for the Promotion of Science (JSPS). Funder website: https://kaken.nii.ac.jp/en/grant/KAKENHI-PROJECT-22K13965/. The funders had no role in study design, data collection and analysis, decision to publish, or preparation of the manuscript.

**Competing interests:** The authors have declared that no competing interests exist.

a random variable $\epsilon$, which may be normal, then transmission of noisy information about the signal can be modelled in the form of learning the value of the information variable

$$\xi = \sigma X + \epsilon, \tag{1}$$

where the parameter $\sigma > 0$ represents the signal-to-noise ratio of this message. For example, the signal $X$ may represent a set of $n$ alternatives labelled by $x_1, x_2, \cdots, x_n$, when the receiver of the message wishes to identify the most appropriate choice out of the $n$ alternatives, and seeks partial information to improve their decision. Communication theory tells us that the simplest form of modelling information acquisition of this kind is to make use of the information variable (1). This information is partial because there are two unknowns for the receiver, the signal $X$ and the noise $\epsilon$, and only one known, the information variable $\xi$. Thus acquiring the information $\xi$ is insufficient to determine the value of $X$, but it is sufficient to reduce its uncertainty [10].

Before acquiring information, that is, before the detection of $\xi$, the prior view of the receiver of the information about the choice $X$ is represented by the probabilities $\{p_k\}$ that $X$ takes the values $\{x_k\}$. If $p(y)$ is the density function for the noise variable $\epsilon$, then after the detection of $\xi$, the perception of the receiver is updated according to the Bayes formula

$$p_k \to \pi_k(\xi) = \frac{p_k \, p(\xi - \sigma x_k)}{\sum_k p_k \, p(\xi - \sigma x_k)}. \tag{2}$$

In this way we can model the transformation $p_k \to \pi_k(\xi)$ of the perception of people on a given subject, based on the arrival of noisy information $\xi$.

In this scheme of modelling the processing of noisy information using the framework of communication theory, there are two canonical ways, passive and active, to model disinformation in a message. The first is to modify the signal-to-noise ratio $\sigma$ in such a way that the receiver is unaware of the modification [12]. This can be achieved either by secretly introducing more noise and confusion to reduce $\sigma$, or secretly releasing more reliable information to increase $\sigma$. In either case, if the receiver is under the impression that the detected information is of the form $\xi = \sigma X + \epsilon$, while in reality it is $\xi' = \sigma' X + \epsilon$, then the resulting inference will be skewed. For instance, if $\sigma' < \sigma$, then the receiver is overconfident about the information obtained, and conversely if $\sigma' > \sigma$, then the receiver is under-confident about the information. We say that this form of modelling disinformation is "passive" because it does not involve an active dissemination of biased and unsubstantiated views. Rather, it merely attempts to alter the level of confusion.

An active form of disinformation, on the other hand, can be modelled by including a bias in the noise [13]. To this end we remark that a genuine noise by definition has no unknown bias. That is, the unconditional expectation of noise in effect vanishes so that $\mathbb{E}[\epsilon] = 0$, because any known bias in $\epsilon$ would be discarded by the receiver anyhow. Hence if the information takes the form

$$\xi = \sigma X + (\epsilon + f), \tag{3}$$

where $f$ is independent of $X$ and has nonzero mean so that $\mathbb{E}[f] \neq 0$, and if in addition the receiver of this information is unaware of the existence of the noise bias $f$, then their assessment of this information is distorted. Specifically, the net effect is to shift the signal

according to

$$X \to X + f/\sigma. \tag{4}$$

Thus, if the disseminator of disinformation wishes the receiver to be misled in thinking that the alternatives labelled by smaller values of $x_k$ be more likely, then it suffices to let $f < 0$. Conversely, a positive $f$ leads to an assessment whereby the receiver thinks those alternatives labelled by larger values of $x_k$ more likely.

In more specific terms, because the receiver by assumption is unaware of the existence of $f$, their assessment is reflected in the view modelled by $\{\pi_k(\xi)\}$. However, in reality the value of $\xi$ has been shifted by $f$, implying that their assessment is skewed into a modified view $\{\pi_k(\xi + f)\}$. In Fig 1 we sketch examples of how the effect of disinformation manifests itself in such an assessment. Starting with a uniform prior over five alternatives, a detection of the message $\xi$ turns this into the corresponding posterior view, but this is skewed towards one direction or the other, depending on whether $f$ is positive or negative.

We see therefore that we arrive at a simple modelling framework that captures people's responses to disinformation, entirely within the framework of elementary communication theory. An important point to keep in mind is that the receiver is unaware of the existence of the noise bias $f$. This is in keeping with the fact that if a receiver of a message knows that it contains disinformation, that is, if the receiver knows the existence and the value of $f$, then that part of the message will be discarded, if the objective is to make the most informed choice.

This active way of modelling disinformation was first envisaged in [13] and it has been applied to investigate the impact of disinformation in electoral competitions [14,15]. The purpose of the present paper is to investigate the effect of combining different information sources, when one, or some of these sources contain disinformation. Specifically, the present paper is organised as follows. We first consider how two information sources on a given topic combine and interact. The result shows that depending on the relative strengths of the signal-to-noise ratios, the intended impact of a given information source can generate an opposite effect to a receiver who consumes both information sources. This means, in particular, that, should there be any disinformation contained in that information source, it will backfire. That

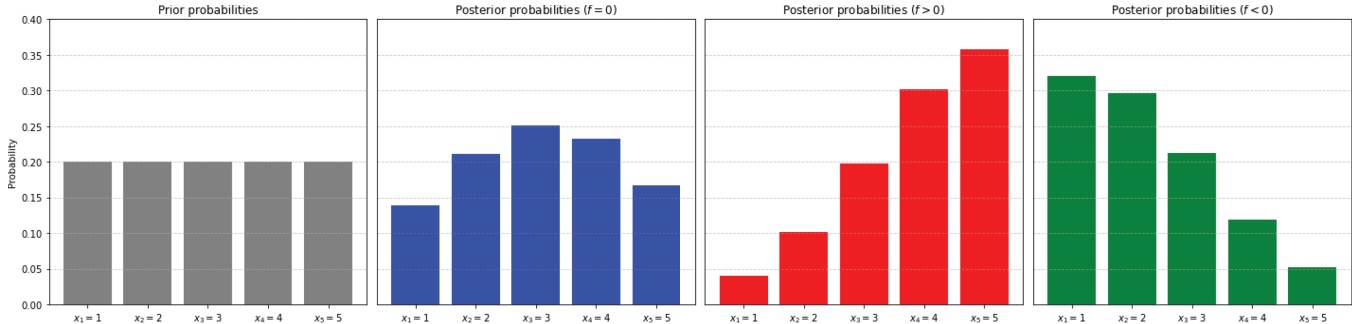

**Fig 1. Perceptions skewed by disinformation**. On the left panel the prior probabilities $\{p_k\}$, here taken to be uniform, for the random variable $X$ representing five possible alternatives, are plotted in the form of a histogram. Having sampled the value of $\xi = \sigma X + \epsilon$, where the signal-to-noise ratio is taken to be $\sigma = 0.5$ and $\epsilon$ is a zero-mean normal variable with standard deviation 1, the prior transforms into posterior $\{\pi_k(\xi)\}$ (the second panel). However, if the message $\xi$ is contaminated with disinformation with $f = 1 > 0$, whose existence is unknown to the receiver, then the posterior view is skewed to the right (the third panel); whereas if $f = -1 < 0$ then it is skewed to the left. The sampled value of the information is $\xi \approx 1.59$.

is, fake news will turn against its spreader in that scenario. We consider a time series generalisation of the model to capture circumstances involving a continuous arrival of information, and illustrate the effect in that setup. We then examine another phenomenon, not related to the issue on how information sources are combined, but nonetheless illustrates circumstances in which an excessive dissemination of disinformation can result in an unintended outcome for the disseminator of the disinformation. We conclude the paper with a brief discussion on broader issues.

Before we proceed, however, we remark on the basic assumption underpinning our analysis, namely, that people's views are updated in accordance with the Bayes formula, for example, as in Eq (2). In cognitive science, adherence to Bayesian updating is viewed as a model for rational behaviours. Yet, there are well-documented violations of the Bayesian updating that have been observed empirically. These violations can broadly be classified into two categories, namely, an apparent violation due to the lack of precise mathematical models to characterise the observed phenomena, and a fundamental violation due to the inapplicability of the Bayes formula to those phenomena. An example of the former is the phenomenon known as "belief polarisation", whereby the opinions held by two people diverge after they have consumed same information [16]. It has been shown that when a detailed Bayesian analysis is applied, such a phenomenon can emerge from an entirely rational behaviour [15,17,18], hence justifying the use of the Bayes formula after all. Examples of the latter include phenomena known as "conjunction fallacy" [19], "disjunction effect" [20], and "question-order effect" [21]. These effects concern propositions and sentiments that are incompatible in the sense that there are no joint probability distributions to simultaneously model different sentiments involved in these effects [22]. In these situations, owing to the lack of joint densities, the Bayes formula is simply inapplicable. In other words, it is not the case that the Bayes formula is violated in the usual sense of violation. Rather, one has to apply a mathematical framework that extends the Bayes formula that is applicable to situations involving incompatible sentiments [22]. When this is done, many of these phenomena can be seen as arising from rational decisions [23]. We shall not further discuss this matter because it falls outside of our consideration here, although it suffices to say that our approach extends naturally to these generalised mathematical framework, and hence our conclusions will not be affected by the existence of these phenomena.

## Combining different information sources

In general there can be several information sources for a given topic. To model this situation, let us consider first the case in which there are two information sources:

$$\xi_1 = \sigma_1 X + \epsilon_1 \quad \text{and} \quad \xi_2 = \sigma_2 X + \epsilon_2 \,. \tag{5}$$

For now we assume that these information sources are free of disinformation. For definiteness we assume that $\epsilon_1$ and $\epsilon_2$ are both zero-mean normal random variables with the same standard deviation, having the correlation $\rho$. Note that the assumption $\mathbb{E}[\epsilon_1^2] = \mathbb{E}[\epsilon_2^2]$ here on the standard deviations of the two noise terms yields no loss of generality because if the standard deviations were different, then we can make them the same by rescaling the signal-to-noise ratio.

The idea represented in (5) is that the receiver gets two messages on the same topic, modelled by $\xi_1$ and $\xi_2$. We shall now show that the aggregate of these messages can be expressed in the form of a single message of the form (1).

We begin by remarking that because the correlation of $\epsilon_1$ and $\epsilon_2$ is $\rho$, there is a zero-mean normal variable $\bar{\epsilon}$, independent of $\epsilon_1$, such that we can write

$$\epsilon_2 = \rho\,\epsilon_1 + \sqrt{1-\rho^2}\,\bar{\epsilon}\,. \tag{6}$$

Analogously, we can define an auxiliary information variable $\bar{\xi}$ by the relation

$$\xi_2 = \rho\,\xi_1 + \sqrt{1-\rho^2}\,\bar{\xi}\,. \tag{7}$$

By a rearrangement of terms we thus have

$$\bar{\xi} = \frac{\sigma_2 - \rho\sigma_1}{\sqrt{1-\rho^2}}\,X + \frac{\epsilon_2 - \rho\epsilon_1}{\sqrt{1-\rho^2}} = \bar{\sigma}\,X + \bar{\epsilon}\,, \tag{8}$$

where $\bar{\sigma}$ is defined to be the coefficient of $X$ in the middle expression. It is evident from (7) that the specification of the two messages $\xi_1$ and $\xi_2$ is equivalent to that of $\xi_1$ and $\bar{\xi}$, however, the latter two messages have independent noise terms $\epsilon_1$ and $\bar{\epsilon}$. If we define $\delta$ according to

$$\delta = \frac{\xi_1}{\sigma_1} - \frac{\bar{\xi}}{\bar{\sigma}} = \frac{\epsilon_1}{\sigma_1} - \frac{\bar{\epsilon}}{\bar{\sigma}}\,, \tag{9}$$

then $\delta$ is independent of $X$, thus represents pure noise. We now construct another zero-mean normal noise variable $\epsilon$ with the properties that it is independent of $\delta$ and that it has the same standard deviation as $\epsilon_1$ and $\epsilon_2$. Writing $\epsilon = \alpha\epsilon_1 + \beta\epsilon_2$, the independence condition $\mathbb{E}[\epsilon\delta] = 0$ shows that $\alpha = \lambda(\sigma_1 - \rho\sigma_2)$ and $\beta = \lambda(\sigma_2 - \rho\sigma_1)$ for some $\lambda$. The variance condition $\mathbb{E}[\epsilon^2] = \mathbb{E}[\epsilon_1^2]$ then fixes $\lambda$ to give

$$\epsilon = \sqrt{\frac{1-\rho^2}{\sigma_1^2 - 2\rho\sigma_1\sigma_2 + \sigma_2^2}}\left[\frac{\sigma_1 - \rho\sigma_2}{1-\rho^2}\,\epsilon_1 + \frac{\sigma_2 - \rho\sigma_1}{1-\rho^2}\,\epsilon_2\right], \tag{10}$$

which, recall, is independent of $\delta$.

To proceed it will be useful to define a parameter $\sigma$ according to

$$\sigma = \sqrt{\frac{\sigma_1^2 - 2\rho\sigma_1\sigma_2 + \sigma_2^2}{1-\rho^2}}\,. \tag{11}$$

In line with (10), and using (11), we now define a new message $\xi$ by setting

$$\xi = \frac{\sigma_1 - \rho\sigma_2}{\sigma(1-\rho^2)}\,\xi_1 + \frac{\sigma_2 - \rho\sigma_1}{\sigma(1-\rho^2)}\,\xi_2\,. \tag{12}$$

Then a direct substitution shows that $\xi$ can be expressed alternatively in the form (1), where $\sigma$ is defined by (11) and $\epsilon$ is defined by (10). Importantly, both $X$ and $\epsilon$ are independent of $\delta$. Further, the specification of the original pair of messages $\xi_1$ and $\xi_2$ is *equivalent* to that of the pair $\xi$ and $\delta$, because there is an invertible linear transformation between these message pairs:

$$\begin{pmatrix} \xi \\ \delta \end{pmatrix} = \begin{pmatrix} \frac{\sigma_1 - \rho\sigma_2}{\sigma(1-\rho^2)} & \frac{\sigma_2 - \rho\sigma_1}{\sigma(1-\rho^2)} \\ \frac{\sigma_2}{\sigma_1(\sigma_2 - \rho\sigma_1)} & -\frac{1}{\sigma_2 - \rho\sigma_1} \end{pmatrix} \begin{pmatrix} \xi_1 \\ \xi_2 \end{pmatrix}. \tag{13}$$

However, because $\delta$ is independent of $X$ and $\epsilon$, knowledge of $\delta$ makes no contribution towards the assessment of the choice represented by $X$. In other words, without loss we can discard $\delta$. It follows that the two messages of (5) can be modelled in the form of a single message of the same form (1). In fact the same conclusion follows even if there are more than two information sources; they will combine to be modelled by a single effective information source (1). Hence we can interpret (1) as representing the aggregate of all messages on the topic modelled by $X$.

With the result (12) at hand we are in the position to assess the impact of disinformation when messages are combined. Specifically, suppose that the first message contained disinformation $f_1$ and the second $f_2$. Then from (12) we deduce that when the two messages are combined, the net effect is to generate a shift

$$\xi \to \xi + \frac{\sigma_1 - \rho\sigma_2}{\sigma(1 - \rho^2)}f_1 + \frac{\sigma_2 - \rho\sigma_1}{\sigma(1 - \rho^2)}f_2\,. \tag{14}$$

If the correlation $\rho$ of the two noise terms $\epsilon_1$ and $\epsilon_2$ is negative, then there are no surprises here, whereas if $\rho > 0$, then there can be an unexpected effect of disinformation. It suffices to consider the contribution of, say, $f_1$. From the form $\xi_1 = \sigma_1(X + f_1/\sigma_1) + \epsilon_1$ of the message we see that a positive $f_1$ is intended to mislead the receiver in thinking that the alternatives $x_k$ with larger values of $k$ more likely than what they actually are. Conversely, if $f_1 < 0$ then the intention is to mislead the receiver in thinking that those alternatives with smaller values of $k$ are more plausible. This is the clear effect of $f_1$ on $\xi_1$, and similarly for $f_2$ on $\xi_2$. However, when two messages are combined, the effect of $f_1$ and $f_2$ on the aggregate information $\xi$ is to generate a shift

$$X \to X + \frac{\sigma_1 - \rho\sigma_2}{\sigma^2(1 - \rho^2)}f_1 + \frac{\sigma_2 - \rho\sigma_1}{\sigma^2(1 - \rho^2)}f_2 \tag{15}$$

of the signal. Observe that when $\rho > 0$, it is possible that one of the coefficients of $f_1$ and $f_2$ be negative (they can be both positive but they cannot both be negative). If that were the case, then the unintended consequence is that the disinformation will generate an opposite effect.

As an example, suppose that $\xi_1$ represents a genuine, albeit noisy, information, while $\xi_2$ contains a deliberate disinformation $f_2$. In this case, if $\sigma_2 < \rho\sigma_1$, then any such disinformation contained in $\xi_2$ will generate a contrary effect on the receiver of both information sources $\xi_1$ and $\xi_2$. We can think, for instance, of a democratic process, such as an election, for which the signal $X$ labels different candidates. In the simplest electoral model setup, the numerical values of the gaps $x_k - x_l$ can be interpreted as representing the relative differences of the positions of the candidates (or their political parties) on the political spectrum [24]. (While more detailed structural models taking into account the various factors that affect voter concerns as well as voter demographics are available [13], the simplest reduced-form model considered here is sufficient to capture qualitative features of the information-based model we wish to highlight here.) Under the convention that $x_n = \max(x_k)$ labels the position that is the farthest to the right and similarly $x_1 = \min(x_k)$ labelling the farthest to the left, a far-right political party naturally wishes the electorates to make the choice $X = x_n$, and similarly for the far-left wishing the choice $X = x_1$. A recent study [25] suggests that political parties at the right-end of the spectrum are significantly more likely to disseminate disinformation. In such a context we can think of $\xi_1$ representing information emanating from centre/left-leaning parties, and $\xi_2$ from the right-leaning parties. Any disinformation contained in $\xi_2$ will thus have the property that $f_2 > 0$.

When the condition $\sigma_2 < \rho\sigma_1$ is met, the impact of such disinformation on the electorates who consume both information sources (and who is unaware of the existence of disinformation) is that the stronger the disinformation is, the less likely it is that they will vote for right-leaning candidates – in spite of the fact that they are unaware of the existence of disinformation. As a strategy to counter the impact of disinformation, therefore, it suffices to increase the value of $\sigma_1$, provided that $\rho > 0$. Intuitively speaking, what this means is that rather than investing energy into denying disinformation, it is more effective simply to release clear and strong message. In Fig 2 we show the statistics on how the impact of the disinformation term in one of the information sources skews the posterior distributions in the intended direction when $\sigma_2 > \rho\sigma_1$, but how this is reversed when the condition $\sigma_2 < \rho\sigma_1$ is satisfied.

Needless to add, such a strategy does not affect those in the so-called disinformation bubble – for instance those who only consume messages represented by $\xi_2$. Also, the positive correlation condition $\rho > 0$ is in general not enforceable by a single information source. Recall that $\rho$ is the correlation between $\epsilon_1$ and $\epsilon_2$; the two noise terms, where by noise we mean rumours, speculations, ambiguities, misinterpretations, and the like. The nature of noise is typically dependent on the communication channel—for example, the media or the internet—and therefore a single information source cannot unanimously fix the value of $\rho$. Nevertheless, when the condition $\sigma_2 < \rho\sigma_1$ is met, the strategy suggested here will be highly effective to counter the impact of disinformation, at least based on what communication theory tells us.

A situation in which the conditions are likely to be met would be for those who consume information delivered through public broadcasters. A public broadcaster tends to cover both sides of the story with the view towards maintaining impartiality. Additionally, the fact that both information sources have gone through the same information channel makes the positivity of noise correlation likely. More generally, we believe, intuitively, but without empirical or scientific evidence, that noise correlations are typically positive (rumours and speculations that obscure one information source will obscure another in a similar manner), based on the intuition that noise is generated independently of the signal by the medium, or the channel, that transports the message.

Before we proceed further, it will be useful to discuss the intuition behind the surprising fact that when $\sigma_2 < \rho\sigma_1$, any disinformation $f_2$ will generate an unintended effect for the disseminator of disinformation. While the mathematical analysis leading to the result (15) is very simple, the effect is nevertheless counterintuitive. To understand this phenomenon, recall that the two messages take the form $\xi_1 = \sigma_1 X + \epsilon_1$ and $\xi_2 = \sigma_2 X + \epsilon_2 + f_2$, where the existence of the term $f_2$ is unknown to the receiver. Now suppose that the strength of the message $\xi_1$ is sufficiently large when compared to that of $\xi_2$, i.e. $\sigma_1 > \sigma_2/\rho$. Suppose also that $f_2$ takes on a large positive value. The intention therefore is to mislead the receiver in thinking that $X$ should take a large value, i.e. to enhance the chance of the receiver selecting the alternatives labelled by larger values of $x_k$. This means that $\xi_2$ will take on a large value, while $\xi_1$ will not. However, the signal strength of $\xi_1$ is strong, so logically, without the knowledge of the existence of $f_2$, the fact that $\xi_2$ is large implies to the receiver that the associated noise $\epsilon_2$ must have taken on a large value. Because $\epsilon_1$ is positively correlated to $\epsilon_2$, it is likely therefore that $\epsilon_1$ must have taken a large value, but $\xi_1$ did not take a large value. It follows, therefore, that $X$ must have take on a small value, not large—contrary to the intention of the disinformation. This is the logic underpinning the phenomenon uncovered here.

## Information flow in continuous time

In the previous section we examined how a rational receiver would respond to a message, or a set of messages, at a single moment in time. In reality people consume information regularly,

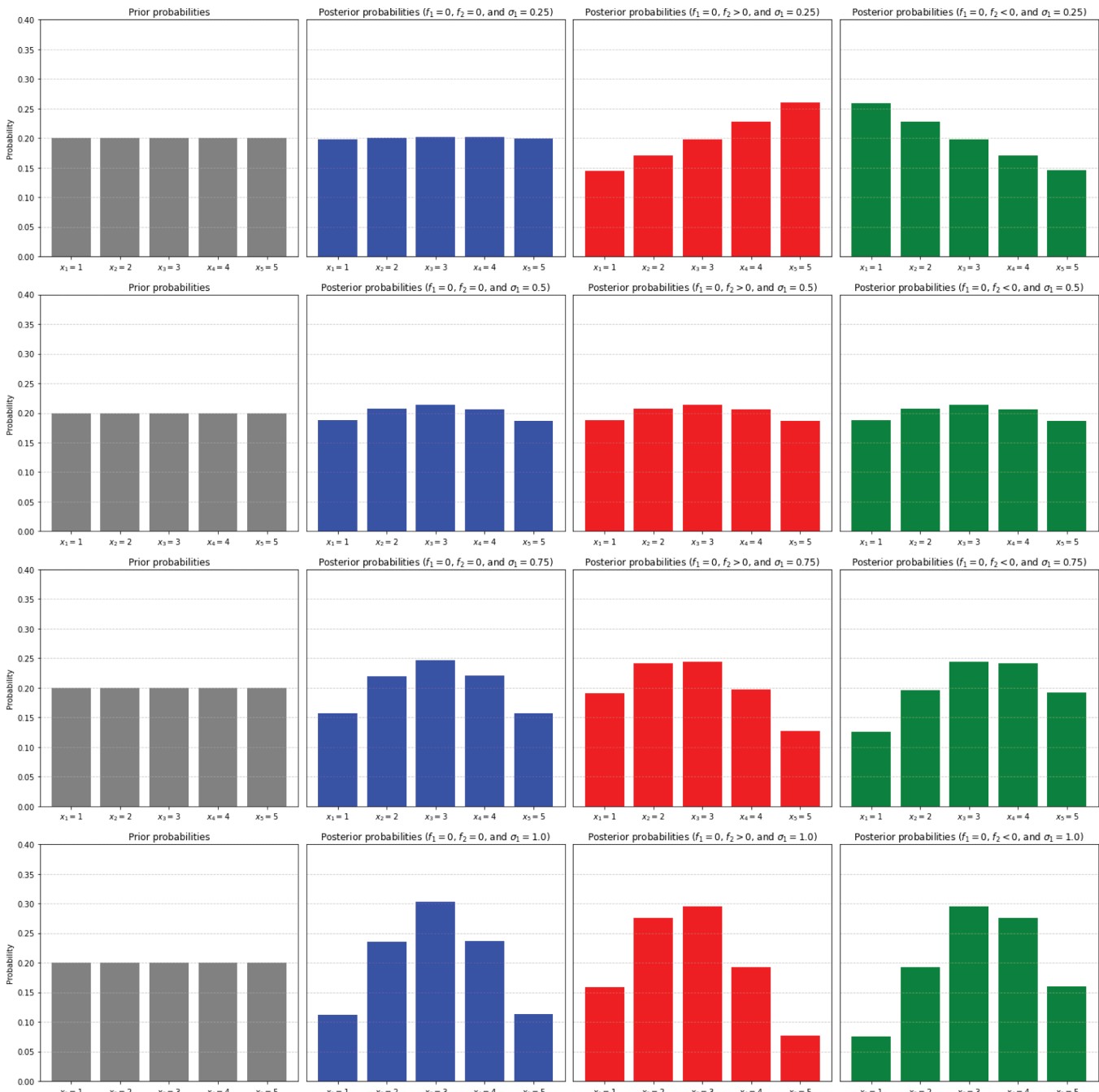

**Fig 2. Skewing and reversal of perceptions by disinformation.** These figures show the expected posterior probabilities $\mathbb{E}[\pi_k|X = x_3]$ based on $10^5$ Monte Carlo trials for the noise term, when the "correct" choice is taken to be $X = x_3$ and when there are two information sources interacting. In all cases, the prior probability (left column, in grey) is uniform, and the parameters are set to $\sigma_2 = 0.25$ and $\rho = 0.5$, while the choice labels are taken to be $x_k = k$ for $k = 1, 2, \ldots, 5$. The conditional averages of the posterior probabilities in the absence of disinformation are shown in the second column (blue) for different values of $\sigma_1 = 0.25, 0.5, 0.75, 1.0$ (top to bottom). In the presence of disinformation in $\xi_2$ with $f_2 = 1.0 > 0$, its impact will be an intended one to skew the posterior to the right (top row in red), while the choice $f_2 = -1.0 < 0$ will skew it to the left (top raw in green), when $\sigma_1 = 0.25$ is insufficiently strong so that $\sigma_2 - \rho\sigma_1 > 0$. When the signal strength in $\xi_1$ is slightly increased to $\sigma_1 = 0.5$ such that $\sigma_2 - \rho\sigma_1 = 0$, the impact of disinformation is exactly cancelled (second row). When the signal strength is increased to $\sigma_1 = 0.75$, we see that the impact of disinformation is reversed (third row). When $\sigma_1$ is further increased to $\sigma_1 = 1.0$, the reversal effect becomes stronger, but at the same time the signal strength is also amplified to enhance the likelihood of the correct choice $X = x_3$ being made (bottom row).

so we wish to extend the 'single-shot' information model (1) into a form of a time series. In general, the quantity of interest, or the signal, the noise, and the signal-to-noise ratio can all vary in time. However, here we consider the case in which the signal remains to be modelled by a fixed random variable $X$ but noise takes the form of a time series $\{\epsilon_t\}$. The rationale for considering a fixed signal is that when considering the impact of disinformation in an electoral competition, which is our primary interest here, it is relatively rare that a candidate is replaced in the middle of an election cycle. If the signal-to-noise ratio is given by a time series $\{\sigma_t\}$, then the time series representing the arrival of information now takes the form

$$\xi_t = X \int_0^t \sigma_s \, \mathrm{d}s + \epsilon_t. \tag{16}$$

This can be interpreted as the time-series version of the simple model (1), when the signal is fixed in time. Because we are working in the context of Gaussian-noise models, it suffices to assume that $\{\epsilon_t\}$ is a standard Brownian motion, which is a Gaussian process with mean zero and standard deviation $\sqrt{t}$, having statistically independent increments.

As before, the signal $X$ represents the choices to be made by the receiver of the information, wherein different alternatives will be modelled by the numbers $\{x_k\}$. The *a priori* probability that $X = x_k$, that is, the probability that the $k$th alternative being selected, is given by $p_k$. For simplicity of exposition, let us for now assume that the signal-to-noise ratio, or the information flow rate, is constant in time so that $\sigma_t = \sigma$. Then the time series representing the message takes on a simple form $\xi_t = \sigma X t + \epsilon_t$. Having detected the message up to time $t$, a rational receiver of the message will update their views on the different alternatives in accordance with the Bayes formula

$$p_k \to \pi_k(\xi_t, t) = \frac{p_k \, e^{\sigma x_k \xi_t - \frac{1}{2}\sigma^2 x_k^2 t}}{\sum_k p_k \, e^{\sigma x_k \xi_t - \frac{1}{2}\sigma^2 x_k^2 t}}, \tag{17}$$

which, in the context of signal processing is also known as the Wonham filer [26]. The fact that the posterior probabilities are functions of $\xi_t$ but not on its history is a consequence of having a time-independent signal-to-noise ratio, making $\xi_t = \sigma X t + \epsilon_t$ a Markov process. For a time-dependent signal-to-noise ratio, on the other hand, the posterior probabilities will be dependent on the entire history of $\{\xi_t\}$ up to time $t$ [14].

We examine now the case in which there are two information-providing time series

$$\xi_t^1 = \sigma_1 X t + \epsilon_t^1 \quad \text{and} \quad \xi_t^2 = \sigma_2 X t + \epsilon_t^2, \tag{18}$$

where the Brownian motions $\{\epsilon_t^1, \epsilon_t^2\}$ are independent of the signal $X$ and have the correlation $\rho$. Then the analysis leading to Eq (12) remains valid and we find that the information contents of the two messages in (18) can be combined into a single information process $\xi_t = \sigma X t + \epsilon_t$, where $\sigma$ is given by (11) and $\epsilon_t$ is given by (10), with $\epsilon_1$ and $\epsilon_2$ replaced by $\epsilon_t^1$ and $\epsilon_t^2$. In particular, if the two messages were to contain disinformation terms $f_t^1$ and $f_t^2$, for instance noise-biasing drifts of the form $f_t^1 = \mu_1 t$ and $f_t^2 = \mu_2 t$, then this will result in shifting the combined information according to

$$\xi_t \to \xi_t + \frac{\sigma_1 - \rho\sigma_2}{\sigma(1 - \rho^2)} \mu_1 t + \frac{\sigma_2 - \rho\sigma_1}{\sigma(1 - \rho^2)} \mu_2 t. \tag{19}$$

Hence as in the previous example, if $\sigma_1 < \rho\sigma_2$ then any intentional disinformation in the message $\xi_t^1$ will work against its spreader, and similarly if $\sigma_2 < \rho\sigma_1$ then any intentional disinformation in the message $\xi_t^2$ will backfire.

As an example, consider the case in which $f_t^1 = 0$ and $f_t^2 = \mu(t - \tau)\mathbb{1}\{t > \tau\}$, where $\mathbb{1}\{t > \tau\}$ denotes the indicator function so that $\mathbb{1}\{t > \tau\} = 0$ if $t \leq \tau$ and $\mathbb{1}\{t > \tau\} = 1$ if $t > \tau$. In other words, at a random time $\tau$, disinformation is released by the second information source. In this case, once the disinformation is released (at time $\tau$), the effective signal modification is given by

$$X \to \tilde{X} = X + \frac{\sigma_2 - \rho\sigma_1}{\sigma^2(1 - \rho^2)} \mu \left(1 - \frac{\tau}{t}\right). \tag{20}$$

The sample paths illustrating the behaviours of posterior probabilities are shown in Fig 3. It is evident that when the signal strength $\sigma_1$ is small, the disinformation will dominate. However, when $\sigma_1$ is increased just enough to ensure $\sigma_2 < \rho\sigma_1$, the effect of disinformation is reversed. An interesting question in this connection concerns the asymptotic (long time) behaviours of these sample paths. For this purpose, we have taken the average over the sample paths for a longer time period, shown in Fig 4. The result confirms the intuition that asymptotically, the choice will converge to the value of $x_k$ that is the closest to the value of $\tilde{X}$. In particular, if $\sigma_1$ is increased sufficiently large, then the reversal effect will also by diminished and we have $\tilde{X} \approx X$, and hence the correct choice will be made over a longer time horizon. But this happens only by sufficiently strengthening the signal strength by those who do not disseminate disinformation.

So far we have considered the case involving two major information sources. When there are multiple information sources, their combined effect on a receiver who consumes all of them is more subtle. Specifically, suppose that there are $n$ information sources on $X$ of the form (18). Then writing $\rho_{ij}$ for the correlation of the noise terms $\epsilon_t^i$ and $\epsilon_t^j$, the aggregate of these $n$ information sources is equivalent to the consumption of a single information

$$\xi_t = \frac{1}{\sigma} \sum_{i,j} \sigma_i \, \rho_{ij}^{-1} \, \xi_t^j, \tag{21}$$

where $\rho_{ij}^{-1}$ denotes the elements of the inverse correlation matrix and

$$\sigma^2 = \sum_{i,j} \sigma_i \, \rho_{ij}^{-1} \, \sigma_j. \tag{22}$$

Therefore, with the information on noise correlation it is possible to identify the impacts of various disinformation terms. However, in practice the estimation of the noise correlation in a multi-media context will be challenging. Nevertheless it is entirely possible that for a fixed $j$ the sum $\sum_i \sigma_i \rho_{ij}^{-1}$ becomes negative, and in this case any disinformation that might be contained in the information source $\xi_t^j$ with an intent to manipulate people in one direction will result in the opposite direction.

## The overshoot effect

We now turn to another unintended consequence that might arise as a result of disseminating disinformation, quite distinct from the reversal effect discussed in the foregoing discussion. To this end we note that the numerical values $\{x_k\}$ that the random variable $X$ takes represent, in the context of an electoral competition, the policy positions of the different candidates

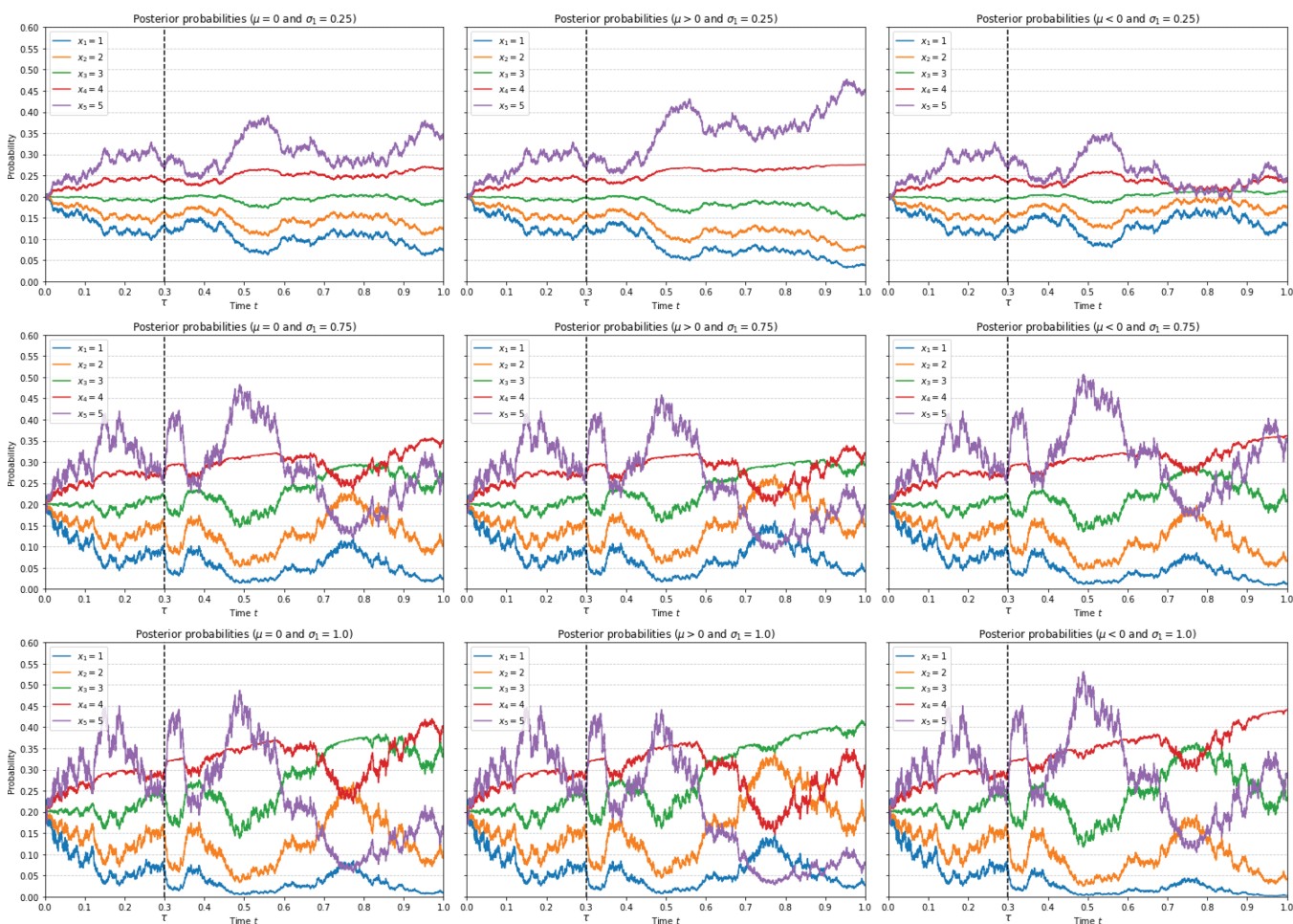

**Fig 3. Sample paths showing the reversal of the impact of disinformation.** On the left column the posterior probabilities $\{\pi_k\}$, conditional on the correct choice being $X = x_3$, are plotted when there are five alternatives labelled by $x_k = k$, $k = 1, 2, \ldots, 5$, in the absence of disinformation, when there are two information sources $\xi_t^1$ and $\xi_t^2$. The impact of disinformation of the form $f_t^2 = \mu(t - \tau)\mathbb{1}\{t > \tau\}$ in $\xi_t^2$ is shown in the middle column (for $\mu = 2.0 > 0$) and the right column (for $\mu = -2.0 < 0$), with $\tau = 0.3$. In all plots, the parameters are set to $\sigma_2 = 0.25$ and $\rho = 0.5$. When $\sigma_1 = 0.25$ so that $\sigma_2 - \rho\sigma_1 > 0$, the desired effect of disinformation is to increase the likelihoods of the alternatives labelled by higher values of $x_k$ (purple and red) when $\mu > 0$, and similarly when $\mu < 0$ the intention is to increase the likelihoods of the alternatives labelled by lower values of $x_k$ (blue and orange), as shown on the top row. However, when $\sigma_1 = 0.75$ so that $\sigma_2 - \rho\sigma_1 < 0$, the effect is reversed (middle row). Increasing the signal strength of $\xi_t^1$ further to $\sigma_1 = 1.0$, the reversal effect is also strengthened, but at the same time due to the strength of the signal, the likelihood of the correct choice being selected (green) also becomes nontrivial (bottom row).

on an issue modelled by $X$. In the simplest case considered here of having just one issue, we may think of them as representing positions of the candidates, or their political parties, on the political spectrum. Without loss we may assume that $x_1 < x_2 < \cdots < x_n$, where $x_n$ represents the farthest position to the right and $x_1$ the furthest to the left.

The overshoot effect discussed below can occur to candidates labelled by $x_2, x_3, \cdots, x_{n-1}$, that is, those who are not at the end of the spectrum, and it can be explained through the following example. Suppose that the majority of the electorates prefer central positions, and that the political party labelled, say, by $x_{n-1}$ wishes to influence the electorates towards further right by means of a dissemination of disinformation. Then for those who consume the information disseminated by that political party, their preferences will shift to the right, but too

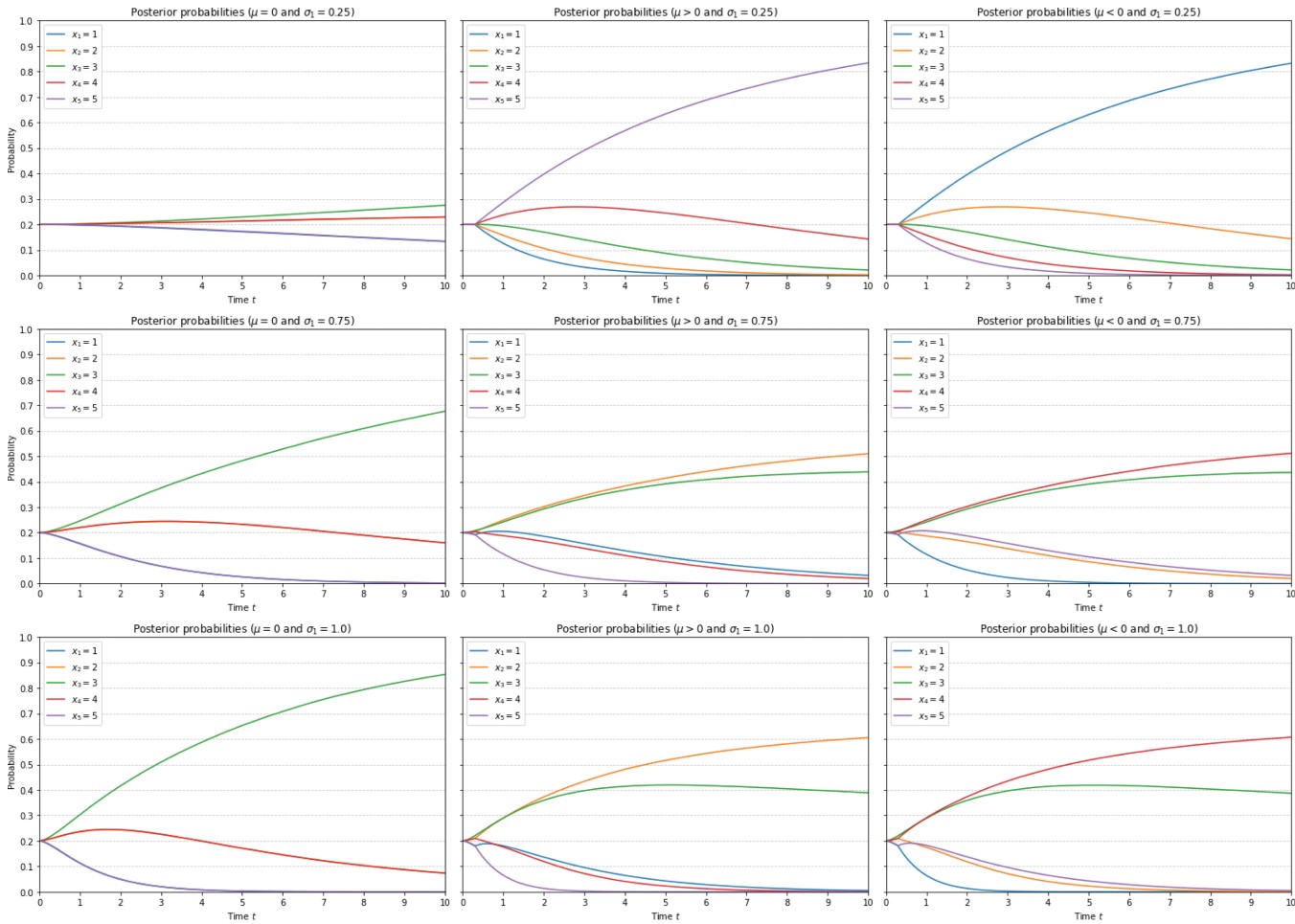

**Fig 4. Averaged sample paths showing the reversal of the impact of disinformation.** An average of $10^5$ sample paths corresponding to each of the plots in Fig 3 is plotted. All parameter values are taken to be the same as those for Fig 3, except here we have extended the time horizon to $T = 10$ to show the asymptotic behaviour of the posterior probabilities. For the parameter values chosen here, we see that the reversal effect will ultimately select the choice $x_2$ if $\mu > 0$ and the choice $x_4$ if $\mu < 0$. Note that for the parameter values used here, the averaged posterior probabilities of $x_1$ and $x_5$ coincide, and likewise those of $x_2$ and $x_4$ coincide, in the absence of disinformation (left column), hence only three curves are visible.

much disinformation will move them too far towards $x_n$, bypassing $x_{n-1}$. This is the essence of the overshoot effect.

At a purely anecdotal level, one can think of the example of the UK Conservative party, the oldest modern political party in the world, over the past five years or so. During this period, in particular leading up to the 2024 general election, political messages from the Conservative government were very much focused on an exaggerated statements about the dangers of migration into the UK. Indeed, "stop the boat" was a catch phrase regularly used like a prayer by the last Conservative prime minister during this period, in reference to the boats carrying undocumented migrants heading towards the UK shores. The reaction of the electorates, however, is to turn to a newly formed Reform UK party, who placed themselves at the far end of the spectrum in relation to their stance against migration. In other words, the messaging probably did have some effects, except that they were unintended ones.

The phenomenon can be understood quite intuitively, but communication theory provides us with both qualitative and quantitative understanding. In essence, the phenomenon is

related to the fact that while $\max_\xi \pi_1(\xi) = \max_\xi \pi_n(\xi) = 1$, $\max_\xi \pi_k(\xi) < 1$ for $k \neq 1, n$. That is, while the posterior probabilities $\pi_1(\xi)$ and $\pi_n(\xi)$ of the extreme ends of the spectrum are unbounded in the range [0,1], those of the intermediate probabilities are bounded by values strictly less than one. To see this, consider the example whereby the noise density is normal with mean zero and variance one. Then we have from the Bayes formula (2) that

$$\pi_k(\xi) = \frac{p_k \, e^{\sigma x_k \xi - \frac{1}{2}\sigma^2 x_k^2}}{\sum_k p_k \, e^{\sigma x_k \xi - \frac{1}{2}\sigma^2 x_k^2}}. \tag{23}$$

It follows at once that

$$\frac{\partial}{\partial \xi} \pi_k = \sigma \pi_k \left( x_k - \sum_k x_k \pi_k \right). \tag{24}$$

Because the expectation $\mathbb{E}[X|\xi] = \sum_k x_k \pi_k$ appearing here is monotonic in $\xi$, it follows that the functions $\pi_k(\xi)$, $k = 2, 3, \ldots, n-1$, have single peaks at the values of $\xi$ for which $\mathbb{E}[X|\xi] = x_k$. It follows that if the value of $\xi$ is too large, perhaps due to an excessive dissemination of disinformation by the candidate labelled, say, by $x_{n-1}$, then the likelihood $\pi_{n-1}(\xi)$ will start to decrease, to be overtaken by the likelihood $\pi_n(\xi)$.

## Discussion

In summary, we have presented two unintended consequences of disseminating disinformation—unintended from the viewpoint of the disseminator. These are concerned with how different information sources combine, and how excessive disinformation can push people away into further extreme positions. Although conditions for which back reactions of disinformation occur cannot always be ensured, when they are met, the theory outlined here on 'information fusion' provides effective measures to counter the impacts of disinformation.

In general, countering the impacts of disinformation is challenging. Fact checkers are useful inasmuch as deciding whether any particular information is genuine or not, but such retrospective analysis have limited effect on countering the impacts. On the other hand, it has been shown that with the statistical information about the disinformation term $f$, a receiver can eradicate the overall majority of the impact of disinformation [13]. That is, even if one does not know whether a given information is true or false, the knowledge of the statistical distribution of disinformation alone is sufficient to eradicate the intended impact of disinformation. In terms of modelling, this corresponds to the situation in which the receiver of a message $\xi = \sigma X + \epsilon + f$ does not know the value of $f$ but is aware of its existence and its distribution. What this means in practice is that data on the statistics of disinformation—evidently available to fact checkers but are rarely made accessible to the public—are more valuable as a tool to counter the overall impacts of disinformation than the outcomes of individual fact checking analysis.

Added to this, in the present analysis we find that when the condition of positive correlation is satisfied (and we have provided an intuitive reasoning why a positive correlation may represent a generic situation), the impact of disinformation can be reversed by simply improving the clarity and signal strength of truthful information. In particular, our analysis suggests that there are limited advantages in devoting much efforts to dispute disinformation; information theory suggests that a clear messaging on the matter of substance on its own is more effective. The clarity of messaging is an important one because electorates generally have aversion against uncertainties [22]. At a purely anecdotal level, the 2024 US presidential

election has perhaps highlighted the effectiveness of the clarity of messaging—for instance, in relation to the ongoing war in Ukraine, the Republican candidate declared that he will end the war within 24 hours of being in office, whereas the Democratic candidate argued that she would stand up to the Russian leader. Irrespective of whether the claims are true or false, the long-term implication of one message appears clear and transparent, while that of the other is highly opaque. More broadly, the general impression was that the messaging of the Republican candidate was assertive, while that of the Democratic candidate was nuanced. Given that electorates desire clarity on the implications of policy positions, the outcome of the election was perhaps not so surprising, at least from the viewpoint of communication theory.

Recent political shifts in the US and elsewhere make the availability of such data less likely. Against this background, any innovation to counter the impacts of disinformation seems valuable towards defending democratic processes, even if their scopes are limited.

## Supporting information

**S1 Dataset. Datasets underlying Figs 1–4.**

- `Fig_1.csv`: Dataset for Fig 1.
- `Fig_2-1.csv`, `Fig_2-2.csv`, `Fig_2-3.csv`, `Fig_2-4.csv`: Datasets for Fig 2 corresponding to $\sigma_1 = 0.25, 0.5, 0.75, 1.0$.
- `Fig_3-1-*.csv`, `Fig_3-2-*.csv`, `Fig_3-3-*.csv`: Datasets for Fig 3 corresponding to $\sigma_1 = 0.25, 0.75, 1.0$, where $* = a, b, c$:
  - a: case shown on the left, no disinformation ($\mu = 0$),
  - b: case shown in the center, positive disinformation ($\mu > 0$),
  - c: case shown on the right, negative disinformation ($\mu < 0$).
- `Fig_4-1-*.csv`, `Fig_4-2-*.csv`, `Fig_4-3-*.csv`: Datasets for Fig 4 corresponding to $\sigma_1 = 0.25, 0.75, 1.0$, following the same * notation as defined for the Fig 3 datasets.

(ZIP)

**S1 Code. Python scripts used to generate Figs 1–4 and S1 Dataset underlying them.**
(ZIP)

## Author contributions

**Conceptualization:** Dorje C. Brody.

**Data curation:** Tomooki Yuasa.

**Formal analysis:** Dorje C. Brody, Tomooki Yuasa.

**Funding acquisition:** Tomooki Yuasa.

**Investigation:** Dorje C. Brody, Tomooki Yuasa.

**Methodology:** Dorje C. Brody, Tomooki Yuasa.

**Project administration:** Dorje C. Brody.

**Software:** Tomooki Yuasa.

**Supervision:** Dorje C. Brody.

**Validation:** Dorje C. Brody, Tomooki Yuasa.

**Visualization:** Tomooki Yuasa.

**Writing – original draft:** Dorje C. Brody.

**Writing – review & editing:** Dorje C. Brody, Tomooki Yuasa.

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
