## [Decision Letter · Decision Letter 0]

11 Jun 2025

PONE-D-25-21796How fake news can turn against its spreaderPLOS ONE

Dear Dr. Yuasa,

Thank you for submitting your manuscript to PLOS ONE. After careful consideration, we feel that it has merit but does not fully meet PLOS ONE’s publication criteria as it currently stands. Therefore, we invite you to submit a revised version of the manuscript that addresses the points raised during the review process.

We look forward to receiving your revised manuscript.

Kind regards,

Carlos Carrasco-Farré

Academic Editor

PLOS ONE

Additional Editor Comments (if provided):

Reviewers' comments:

Reviewer's Responses to Questions

**Comments to the Author**

1. Is the manuscript technically sound, and do the data support the conclusions?

Reviewer #1: Partly

Reviewer #2: Yes

2. Has the statistical analysis been performed appropriately and rigorously? 

Reviewer #1: N/A

Reviewer #2: Yes

3. Have the authors made all data underlying the findings in their manuscript fully available?

Reviewer #1: No

Reviewer #2: Yes

4. Is the manuscript presented in an intelligible fashion and written in standard English?

Reviewer #1: Yes

Reviewer #2: Yes

5. Review Comments to the Author

Reviewer #1: The manuscript highlights a relevant problem regarding fake news management. A crucial point regarding opponent sources of disinformation is treated and, in principle, sounding. The manuscript is interesting and the underlying ideas are sufficiently well explained. However, without numerical experiments or at least a source of validation in a prescribed context makes the analysis of limited value, in my opinion. I strongly suggest to proceed in this direction before re-considering the manuscript for publication in PLOS One.

Reviewer #2: This paper presents a rigorous and conceptually compelling analysis of how disinformation can sometimes have a backfire effect—causing belief updates in the opposite direction to its intended aim. Using tools from communication theory and Bayesian inference, the authors show that when a rational receiver integrates two information sources—one genuine and the other tainted by disinformation—disinformation can be counterproductive if specific conditions are met. In particular, if the genuine source has a higher signal-to-noise ratio and the noise in the two sources is positively correlated, the combined effect may be to shift the receiver's belief away from the disinformation. The authors extend their model to continuous time, generalize it to multiple sources, and identify a second phenomenon—the "overshoot effect"—where excessive disinformation pushes preferences beyond the intended target, potentially toward more extreme alternatives.

1. Positioning in Relation to Prior Literature

The analysis here is distinctive and well-developed, but it exists within a growing literature showing that counterintuitive belief updates can result from rational inference. Two earlier contributions deserve explicit discussion:

Jern, A., Chang, K. M. K., & Kemp, C. (2014). Belief polarization is not always irrational. Psychological Review, 121(2), 206.

Cook, J., & Lewandowsky, S. (2016). Rational irrationality: Modeling climate change belief polarization using Bayesian networks. Topics in Cognitive Science, 8(1), 160–179. https://doi.org/10.1111/tops.12186

These papers demonstrate that belief polarization or divergence can arise in Bayesian agents when they differ in priors or latent worldviews. By contrast, the current paper holds priors fixed and instead focuses on source structure and noise correlation as drivers of reversal effects. This is a meaningful contribution: it isolates a novel channel through which rational agents can arrive at counterintuitive belief shifts—not because of differences in prior beliefs, but due to the structural interplay of multiple noisy inputs. Clarifying this relationship would help establish the paper’s place within this theoretical lineage.

2. Clarification of Model Assumptions

The reversal effect depends on a set of assumptions that, while clearly articulated in the manuscript, could be further examined in terms of their plausibility:

o That receivers are exposed to both high-quality and disinformation-containing messages;

o That they are unaware of the presence of disinformation;

o That the noise components in the two channels are positively correlated.

The authors acknowledge that these conditions may not always be met but could expand on real-world conditions where they are likely to hold—for example, shared narratives across partisan and mainstream news, or algorithmic content blending in social media feeds. A deeper discussion of these boundary conditions would help readers evaluate when and where the theory is most applicable.

3. Consideration of Human Cognitive Constraints

The model assumes a rational Bayesian receiver who integrates messages based on signal-plus-noise structure without bias or selective attention. This is analytically useful but overlooks the many well-established deviations from rationality in human information processing—such as motivated reasoning, source derogation, and confirmation bias. The authors should briefly acknowledge these factors and reflect on how they might interact with or obscure the modelled reversal effect. For instance, would confirmation bias neutralise the backfire effect? Could heuristic discounting of discordant signals block the reversal from emerging? Even a brief engagement with this literature would ground the analysis more firmly in interdisciplinary discussions.

4. Practical and Policy Implications

The findings have immediate relevance for debates on how to counter disinformation. Yet the policy implications are only lightly touched upon. The authors could expand on this by considering:

o Whether the model supports focusing on amplifying high-quality information rather than censoring low-quality content;

o Whether pre-bunking (forewarning) could interact with the reversal effect;

o How this mechanism might inform platform design or media literacy strategies.

The observation that the best counter to disinformation may be simply to improve the clarity and signal strength of truthful information is both powerful and actionable. Developing this thread would increase the paper’s relevance beyond theory.

6. PLOS authors have the option to publish the peer review history of their article (what does this mean?). If published, this will include your full peer review and any attached files.

Reviewer #1: No

Reviewer #2: **Yes: **

---

## [Author Response · Author response to Decision Letter 1]

14 Aug 2025

Response to Referee 1

We thank the referee for their suggestion of enhancing the numerical experiments in the paper. We agree that such an analysis will improve the quality of the paper, and accordingly we have carried out further numerical analysis. Specifically, the single sample of Figure 1 in the original manuscript has been sampled ten thousand times to take the average to show the statistics of the effect identified in our paper. These are shown as Figure 2 in the revised manuscript. The simulated sample paths in Figure 2 of the original manuscript have been expanded with different parameter values, shown in Figure 3 of the revised manuscript. Further, to show the corresponding statistics, ten thousand simulations have been carried out to show the averaged effect, included in Figure 4 of the revised manuscript.

Response to Referee 2

We thank the referee for making a number of interesting and valuable suggestions, all of which we agree would enhance the interest of our paper. Our response to the specific points raised by the referee are as follows.

1. Positioning in relation to prior work. We thank the referee for pointing out these interesting papers that examine belief polarisation effect using Bayesian net and other approaches. In the revised manuscript we have briefly commented on these papers, in the paragraph starting at the bottom of page 3 of the revised manuscript.

2. Clarification of model assumptions. We agree with the referee that, while we have stated the assumptions in our original submission, further remarks on these assumptions would be highly beneficial. In the revised manuscript we have inserted a paragraph commenting on the model parameter conditions, in the paragraph starting towards the bottom of page 6 of the revised manuscript.

3. Consideration of human cognitive constraints. We agree this is an important (and can also be contentious) issue that should be discussed at least briefly. In the revised manuscript we have included a brief discussion on the issue of the use of Bayesian method in our paper, in the paragraph starting at the bottom of page 3. One important point here is that a number of well-documented violations of the Bayesian updating can be explained based on a framework that generalised the Bayes formula, and our approach remains applicable in the broader framework. We have commented briefly on this, although we did not elaborate on it because it is tangential to the effect investigated in the present paper.

4. Practical and policy implications. We agree with the referee that such a discussion would increase the paper’s relevance. In the revised manuscript we have inserted a discussion on this, in the penultimate paragraph of the paper.

---

## [Decision Letter · Decision Letter 1]

19 Aug 2025

How fake news can turn against its spreader

PONE-D-25-21796R1

Dear Dr. Yuasa,

We’re pleased to inform you that your manuscript has been judged scientifically suitable for publication and will be formally accepted for publication once it meets all outstanding technical requirements.

Kind regards,

Carlos Carrasco-Farré

Academic Editor

PLOS ONE

Additional Editor Comments (optional):

Reviewers' comments:

Reviewer's Responses to Questions

**Comments to the Author**

1. If the authors have adequately addressed your comments raised in a previous round of review and you feel that this manuscript is now acceptable for publication, you may indicate that here to bypass the “Comments to the Author” section, enter your conflict of interest statement in the “Confidential to Editor” section, and submit your "Accept" recommendation.

Reviewer #1: All comments have been addressed

Reviewer #2: All comments have been addressed

2. Is the manuscript technically sound, and do the data support the conclusions?

Reviewer #1: (No Response)

Reviewer #2: Yes

3. Has the statistical analysis been performed appropriately and rigorously? 

Reviewer #1: (No Response)

Reviewer #2: Yes

4. Have the authors made all data underlying the findings in their manuscript fully available?

Reviewer #1: (No Response)

Reviewer #2: Yes

5. Is the manuscript presented in an intelligible fashion and written in standard English?

Reviewer #1: (No Response)

Reviewer #2: Yes

6. Review Comments to the Author

Reviewer #1: The authors have addressed my comments. I think that the manuscript has reached a level sufficient for publication.

Reviewer #2: All comments addressed. No further concerns. Nothing further to say.

---

## [Editor Report · Acceptance letter]

PONE-D-25-21796R1

PLOS ONE

Dear Dr. Yuasa,

I'm pleased to inform you that your manuscript has been deemed suitable for publication in PLOS ONE. Congratulations! Your manuscript is now being handed over to our production team.

Kind regards,

on behalf of

Dr. Carlos Carrasco-Farré

Academic Editor

PLOS ONE